# Combination Therapy of Placenta-Derived Mesenchymal Stem Cells with WKYMVm Promotes Hepatic Function in a Rat Model with Hepatic Disease via Vascular Remodeling

**DOI:** 10.3390/cells11020232

**Published:** 2022-01-11

**Authors:** Ji Hye Jun, Sohae Park, Jae Yeon Kim, Ja-Yun Lim, Gyu Tae Park, Jae Ho Kim, Gi Jin Kim

**Affiliations:** 1Department of Biomedical Science, CHA University, Seongnam 13488, Korea; jihyejun1015@gmail.com (J.H.J.); sohae11@snu.ac.kr (S.P.); janejaeyeon92@gmail.com (J.Y.K.); 2Department of Integrated Biomedical and Life Science, College of Health Science, Korea University, Seoul 02841, Korea; jayun78@korea.ac.kr; 3Department of Physiology, School of Medicine, Pusan National University, Yangsan 50612, Korea; daramzuy2@naver.com (G.T.P.); jhkimst@pusan.ac.kr (J.H.K.)

**Keywords:** liver cirrhosis, placenta-derived mesenchymal stem cells, WKYMVm, combination therapy

## Abstract

Changes in the structure and function of blood vessels are important factors that play a primary role in regeneration of injured organs. WKYMVm has been reported as a therapeutic factor that promotes the migration and proliferation of angiogenic cells. Additionally, we previously demonstrated that placenta-derived mesenchymal stem cells (PD-MSCs) induce hepatic regeneration in hepatic failure via antifibrotic effects. Therefore, our objectives were to analyze the combination effect of PD-MSCs and WKYMVm in a rat model with bile duct ligation (BDL) and evaluate their therapeutic mechanism. To analyze the anti-fibrotic and angiogenic effects on liver regeneration, it was analyzed using ELISA, qRT-PCR, Western blot, immunofluorescence, and immunohistochemistry. Collagen accumulation was significantly decreased in PD-MSCs with the WKYMVm combination (Tx+WK) group compared with the nontransplantation (NTx) and PD-MSC-transplanted (Tx) group (*p* < 0.05). Furthermore, the combination of PD-MSCs with WKYMVm significantly promoted hepatic function by increasing hepatocyte proliferation and albumin as well as angiogenesis by activated FPR2 signaling (*p* < 0.05). The combination therapy of PD-MSCs with WKYMVm could be an efficient treatment in hepatic diseases via vascular remodeling. Therefore, the combination therapy of PD-MSCs with WKYMVm could be a new therapeutic strategy in degenerative medicine.

## 1. Introduction

Angiogenesis is essential in many biological processes, including development and wound repair [1]. Progressive hepatic vascular pressure due to repetitive hepatic damage is the major cause of liver cirrhosis [2]. Abnormal hepatocyte–endothelium crosstalk in the injured liver delays regeneration by promoting fibrogenesis and the formation of scar tissues [3]. Additionally, the irregular vascular system not only indicates the fibrotic pathophysiology but also inhibits metabolism in the liver [4]. The hepatic vascular niche, which is mainly composed of liver sinusoidal endothelial cells (LSECs), secretes angiocrine factors such as vascular endothelial growth factor (VEGF), hepatic growth factor (HGF), Wnt, and endoglin (CD105) to promote hepatic regeneration [5,6]. Increased expression of VEGF leads to hepatic regeneration following partial hepatectomy or drug intoxication, and VEGF is a strong modulator of vascular regeneration due to its effects on endothelial cell proliferation and survival [7,8]. Additionally, the secretion of HGF in endothelial cells inhibited the activation of fibroblasts and functioned as a key molecule to prevent fibrosis in a mouse model of liver fibrosis [9]. In endothelial cells, Wnt/β-catenin pathway activation by Wnt ligands can induce cell cycle progression through transcriptional activation of cyclin D1 and regulate endothelial permeability [10,11].

Mesenchymal stem cell (MSC) therapy is a promising strategy for treatment of hepatic diseases through overexpression of HGF or matrix metalloproteinases (MMPs) and a reduction in collagen levels [12,13]. Among MSCs, human placenta-derived MSCs (PD-MSCs), obtained from fetal tissue, have various advantages, such as strong immunosuppressive abilities, multipotent differentiation, and self-renewal properties [14,15,16,17]. Previously, we reported that PD-MSCs have therapeutic effects in a carbon tetrachloride (CCl_4_)-injured rat model via an autophagic mechanism and IL-6/gp130/signal transducer and activator of transcription 3 (STAT3) pathway [18,19]. Additionally, we demonstrated that PD-MSCs enhance hepatic regeneration via restoration of hepatic lipid metabolism and vascular regeneration by microRNAs (miRNAs) in a rat model of bile duct ligation (BDL) [20,21].

The synthetic peptide WKYMVm, which is composed of Trp-Lys-Tyr-Met-Val-D-Met, is a potent agonist of formyl peptide receptor 2 (FPR2), which belongs to the G-protein coupled receptor family [22,23]. FPR2 promotes vessel growth and cell proliferation [23,24]. The therapeutic effects of WKYMVm have been demonstrated in various degenerative diseases. In ischemic hind limb and myocardial diseases, WKYMVm promoted neovascularization and tissue repair through migration and proliferation of endothelial colony-forming cells (ECFCs) or circulating angiogenic cells (CACs) [25,26]. Topical application of WKYMVm intensified cutaneous wound healing in streptozotocin (STZ)-induced diabetes via the remodeling of von Willebrand factor (vWF)-positive vessels [27]. For treatment of bronchopulmonary dysplasia (BPD), a chronic lung disease, WKYMVm was administered through intraperitoneal injection [28]. Additionally, polylactic-co-glycolic acid (PLGA) could form microspheres encapsulating WKYMVm and consistently released peptides following administration to ischemic hind limbs, extending angiogenic stimulation [29].

However, the effects of combined PD-MSCs and WKYMVm in liver cirrhosis have not been elucidated. Therefore, our objectives were to analyze the effects of PD-MSCs with WKYMVm on hepatic regeneration via vascular regeneration in a BDL-injured rat model of hepatic fibrosis. We investigated the mechanism underlying the therapeutic effects in a liver injury model.

## 2. Materials and Methods

### 2.1. Materials

The structure of the WKYMVm peptide was reported by Choi et al. [30]. The synthetic peptide WKYMVm was synthesized at ANYGEN (Kwangju, Korea). The purity of synthesized WKYMVm was >98%.

### 2.2. Cell Culture

Collection of placental samples from healthy women (≥37 gestational weeks) for research purposes was approved by the Institutional Review Board of CHA Gangnam Medical Center, Seoul, Korea (IRB 07-18). PD-MSCs were isolated and maintained as described previously [31]. A rat hepatocyte-like epithelial cell line, WB-F344, and rat T-HSC/Cl-6 cells, rat hepatic stellate cells (HSCs) transformed with simian virus 40, were maintained in α-MEM supplemented with 10% fetal bovine serum (FBS; Gibco, Langley, OK, USA, 16000-044), and 100 U/mL penicillin–streptomycin (Pen-Strep, P/S; Gibco, 15-140-122). Additionally, human umbilical vein endothelial cells (HUVECs) were cultured in endothelial cell medium (ECM; ScienCell, Carlsbad, CA, USA, 1001) under 5% CO_2_ at 37 °C. Furthermore, WB-F344, T-HSC/Cl-6, and HUVECs were treated with lithocholic acid (LCA; 100 μM; Sigma, Burlington, MA, USA, L6250), transforming growth factor-beta (TGF-β; 2 ng/mL; PeproTech, Rocky Hill, NJ, USA, 100-21C), and 5-fluorouracil (5-FU, 1 μg/mL, Sigma, F6627) for 24, 48, and 48 h, respectively. The WKYMVm peptide was administered at a concentration of 1 mM in vitro.

### 2.3. Animals

Seven-week-old male Sprague Dawley (SD) rats (Orient Bio, Inc., Seongnam, Korea) were maintained in an air-conditioned facility. The common bile duct was ligated. Additionally, detailed protocol is described in Appendix A. All animal experimental processes were approved by the protocol consistent with the Institutional Review Board of CHA General Hospital, Seoul, Korea. The experimental protocols were approved by the Institutional Animal Care and Use Committee of CHA University, Seongnam, Korea (IACUC-180023).

### 2.4. Biochemical Analysis

The harvested serum was examined for aspartate aminotransferase (AST), alanine aminotransferase (ALT), total bilirubin, and albumin (ALB) by Southeast Medi-Chem Institute (Busan, Korea). The experiment was performed in triplicate.

### 2.5. Cell Proliferation Assay

To confirm the proliferative effects of WKYMVm on PD-MSCs, HUVECs, and WB-F344, we incubated Cell Counting Kit-8 (CCK-8; Dojindo Molecular Technologies, Inc., Rockville, MD, USA, CK04) solution with cells at 37 °C for 2 h. The absorbance at 450 nm for CCK-8 solution was measured with an epoch microplate spectrophotometer (BioTek, Winooski, VT, USA). All reactions were performed in triplicate.

### 2.6. Senescence-Associated β-Galactosidase (SA-β Gal) Assay

SA-β-gal activity was analyzed using an SA-β-gal staining kit (Cell Signaling, Danvers, MA, USA, 9860S) according to the manufacturer’s instructions. The intensity of SA-β-gal was analyzed with a microscope via a high-magnification digital camera (Nikon Instrument, Nikon, Inc., Melville, NY, USA) and the ImageJ program (NIH).

### 2.7. Differentiation of WKYMVm-Treated PD-MSCs

To evaluate the potential for differentiation, we seeded PD-MSCs at a density of 2 × 10^4^ cells/35 mm plate and maintained them in α-MEM supplemented with 10% FBS, 100 U/mL of Pen/Strep, 25 ng/mL of FGF4 (PeproTech, AF-100-31), and 1 μg/mL of heparin (Sigma, H3149) until they reached 60% confluency. Then, 1 mM of WKYMVm peptide was added to PD-MSCs for 24 h. Adipogenic and osteogenic differentiation was induced by media from the following: StemPro Adipogenesis Differentiation Kit (Gibco) and StemPro Osteogenesis Differentiation Kit (Gibco), respectively. The differentiation media were replaced every 3 days until day 21.

### 2.8. Tube Formation Assay

For analysis of the angiogenic ability of endothelial cells, HUVECs were stained with Alexa Fluor 488 Ac-LDL (Invitrogen, L23380). HUVECs (5 × 10^4^ cells/well) were precoated with Matrigel (Corning Inc., Corning, New York, NY, USA, 354248) in 24-well culture plates, seeded, and cultured with or without 5-FU and 1 mM of WKYMVm and co-cultivated with PD-MSCs at 37 °C in a 5% CO_2_ incubator. The sprouted tube length was measured and quantified using the ImageJ program. All experiments were conducted in triplicate.

### 2.9. Dextran Permeability Assay

HUVECs were seeded in the upper chamber of a 24-well transwell system and cultured for 24 h to allow growth of a confluent monolayer. Monolayers were treated with 100 μg/mL of 5-FU. After 72 h, monolayers were administered with 1 mM of WKYMVm peptide and co-cultivated with PD-MSCs for 24 h. Dextran permeability was tested by adding 10 µL of 10 mg/mL fluorescein isothiocyanate (FITC)-dextran (Sigma) to the upper chamber for 30 min. After 30 min, 100 µL of conditioned medium in the lower chamber was transferred to a 96-well plate (BD Biosciences, Bedford, MA, USA) and read at excitation and emission wavelengths of 490 and 525 nm, respectively. The experiments were performed in triplicate.

### 2.10. Histological Analysis

The liver tissues were fixed in 10% (*v*/*v*) neutral buffered formalin (NBF; BBC Biochemical, Mount Vernon, WA, USA), embedded in paraffin, cut into 5 μm sections and stained with hematoxylin and eosin (H&E) and Sirius Red. The diameter of the hepatic portal vein and collagen deposition area were measured by ImageJ software through H&E and Sirius Red staining, respectively.

### 2.11. Immunohistochemistry

To analyze hepatocyte proliferation in tissues following injection with PD-MSCs and WKYMVm or nontransplantation, we examined the expression of proliferating cell nuclear antigen (PCNA) localized in the nucleus. The sectioned slides were incubated in 3% H_2_O_2_ in methanol to block endogenous peroxidase. After antigen retrieval through microwaving, the slides were reacted with an anti-PCNA antibody (Santa Cruz Biotechnology, Dallas, TX, USA, sc-56) and diluted with antibody diluent (Dako, Santa Clara, CA, USA, S3022) at 4 °C overnight, followed by 30 min with biotinylated secondary anti-rabbit antibody at room temperature (RT). Incubation with horseradish peroxidase-conjugated streptavidin–biotin complex (Dako, K1015) and 3,3-diaminobenzidine (Dako, K1015) was used to induce chromatic signals. The slides were counterstained with Mayer’s hematoxylin (Dako, S-3099). Images were detected using a digital slide scanner (3DHISTECH, Ltd., Budapest, Hungary). Finally, the percentage of PCNA-positive hepatocytes was measured in all sections at 400× magnification.

### 2.12. Immunofluorescence

For analysis of the localization of vWF, α-SMA, active β-catenin, and FPR2 in vivo or in vitro, primary antibodies against vWF (1:200, Abcam, Cambridge, MA, USA, ab6994), α-SMA (1:200; Dako, M0851), active β-catenin (1:100; Cell Signaling Technology, Danvers, MA, USA, 8814S), and FPR2 (1:100; Novus, St. Louis, MO, USA, NLS1878) was added to antibody diluent (Dako, S3022) and reacted with slides at 4 °C overnight. The secondary antibody Alexa Fluor 488 and 594 (1:400; Invitrogen, Camarillo, CA, USA, A21206 and A11012) was added for 1 h. The slides were counterstained with 4,6-diamidino-2-phenylindole (DAPI; Invitrogen, D3571). The images were observed with a confocal microscope (LSM 700; Zeiss, Oberkochen, Germany). The observed images were analyzed with ZEN blue software (Zeiss). The experiments were performed in triplicate.

### 2.13. Enzyme-Linked Immunosorbent Assay (ELISA)

The concentrations of VEGF, HGF, and TGF-β were analyzed by ELISAs. Their concentrations were measured using human VEGF (Abcam, ab100662), rat VEGF (Abcam, ab100786), human HGF (R&D Systems, DHG00B), rat HGF (R&D Systems, MHG00), and human TGF-β (Abcam, ab100647) ELISA kits in strict accordance with the manufacturer’s instructions and detected using a microplate reader (BioTek, Winooski, VT, USA) at 450 nm.

### 2.14. Western Blot

Protein lysates were subjected to dodecyl sulfate polyacrylamide gel electrophoresis (SDS-PAGE), transferred to polyvinylidene difluoride membranes (PVDF; Bio-Rad Laboratories), and then blocked-in blocking buffer for 1 h. The membranes were subsequently incubated with rabbit anti-vimentin (1:2000; Sigma, v4630), anti-Col I (1:1000; Abcam, ab34710), anti-GAPDH (1:3000; Abfrontier, Seoul, Korea, LF-PA0018), anti-ALB (1:1000; Novus, NBP1-32458), anti-cyclin D1 (1:1000; Abfrontier, LF-MA0325), anti-gp130 (1:1000; Abcam, ab202850) anti-pSTAT3 (1:1000; Cell Signaling Technology, 9134s), mouse anti-E-cadherin (1:1000; Cell Signaling Technology, 14472s), anti-α-SMA (1:1000; Dako, M0851), anti-IL-6 (1:1000; Abcam, ab6672), and anti-tSTAT3 (1:1000; Cell Signaling Technology, 9139s) antibodies at 4 °C overnight. After the reaction, the membranes were treated with anti-rabbit IgG, HRP-linked antibody (1:10,000; Cell Signaling Technology, 7074P2), anti-mouse IgG, or HRP-linked antibody (1:5000; Cell Signaling Technology, 7076s)-conjugated secondary antibody or for 1 h at RT. The bands were detected using a Clarity Western ECL kit (Bio-Rad Laboratories, 1705061). Western blotting was performed in triplicate.

### 2.15. Quantitative Reverse Transcription-Polymerase Chain Reaction (qRT-PCR)

Total RNA was extracted from rat liver tissues and TRIzol-treated cells (Ambion, Austin, TX, USA, 16696018). Reverse transcription was performed with 250 ng of total RNA and Superscript III reverse transcriptase (Invitrogen). Complementary DNA (cDNA) was synthesized by PCR. Real-time PCR was performed using SYBR Green Master Mix (Roche, Mannheim, Germany) and a CFX Connect Real-Time System (Bio-Rad Laboratories). The PCR conditions were as follows: denaturation at 95 °C for 15 min and 20 s, followed by 40 cycles of 95 °C for 30 s and annealing at 55~60 °C for 40 s. Extension at 70 °C for 15 min and a final extension at 72 °C for 7 min were performed. Gene expression was normalized to that of GAPDH. The primer sequences are shown in Appendix A. All reactions were performed in at least triplicate.

### 2.16. Gelatin Zymography

The expression of MMP-2/9 was detected by gelatin zymography. Supernatants from PD-MSCs or HSCs were collected and separated on 12% SDS polyacrylamide gels supplemented with 1 mg/mL of gelatin. The separated gels were washed twice for 40 min with renaturation buffer (Bio-Rad Laboratories, Hercules, CA, USA, AB102-401) and incubated overnight at 37 °C in developing buffer containing 50 mM of Tris-HCl (pH 7.4), 0.2 M of NaCl, 5 mM of CaCl2, and 1% Triton X-100. The next day, the gels were stained with 10% acetic acid 40% methanol containing 0.5% Coomassie Brilliant Blue R-250 (Bio-Rad Laboratories, 1610406) for 3 h and destained with destaining buffer containing 10% acetic acid 40% methanol. The density of unstained bands was used to detect enzyme expression. The intensity of the gel bands was measured using the ImageJ software program. All experiments were performed in triplicate.

### 2.17. Statistical Analysis

Data are presented as the mean ± standard error of the mean. Differences between different regions of SBEM were analyzed with GraphPad Prism software, and the statistical methods were used for comparisons between pairs. In summary, datasets with more than two groups were analyzed with one-way ANOVA. Datasets with two groups were analyzed with Student’s *t* test. Significance was defined as *p* < 0.05.

## 3. Results

### 3.1. Characterization of PD-MSCs Combined with WKYMVm

To confirm the PD-MSC characteristics after WKYMVm treatment, we analyzed the morphology and proliferative capacity of PD-MSCs and the expression of stemness-related markers (e.g., octamer-binding transcription factor 4 (Oct4), sex-determining region Y-box 2, (Sox2), and Nanog), germ lineage markers (e.g., neurofilament-68 (NF-68) and alpha fetoprotein (AFP)) and the human leukocyte antigen g (HLA-G). The WKYMVm-treated PD-MSCs exhibited an elongated, spindle-shaped morphology similar to that of untreated PD-MSCs (Figure 1A). The proliferative capacity of the WKYMVm-treated PD-MSCs was significantly increased compared to that of the untreated PD-MSCs, as shown by CCK-8 assays (*p* < 0.05, Figure 1B). Additionally, there was no difference in the expression levels of stemness markers between the two groups (Figure 1C). To confirm the immunophenotypes of the PD-MSCs treated with WKYMVm, we analyzed the cell surface markers by flow cytometry. The WKYMVm-treated PD-MSCs were positive for the expression of MSC markers such as CD13, CD90, CD105, HLA-ABC, and HLA-G. However, they were negative for hematopoietic lineage markers such as CD34, CD45, and HLA-DR (Figure 1D).

To investigate the differentiation potential of PD-MSCs with WKYMVm, we maintained the cells in adipogenic or osteogenic induction media for 21 days. Lipid droplets stained with Oil Red O and adipogenic markers (e.g., adipsin, CFD; peroxisome proliferator-activated receptor gamma, PPARG) were highly expressed at the mRNA level in the differentiated WKYMVm-treated PD-MSCs (*p* < 0.05, Figure 1E–G). Additionally, calcium deposition stained by von Kossa staining was observed in the WKYMVm-treated PD-MSCs and untreated PD-MSCs (Figure 1H). The levels of osteogenic markers (e.g., osteocalcin, BGLAP; collagen type I, COL1A1) were dramatically increased in the differentiated WKYMVm-treated PD-MSCs versus the undifferentiated PD-MSCs (Figure 1I,J). These data suggest that the WKYMVm-treated PD-MSCs maintain characteristics similar to those of naïve PD-MSCs.

### 3.2. WKYMVm Enhances the Effects of PD-MSCs

To identify whether PD-MSCs express FPR2, which is the major receptor of the WKYMVm ligand, we analyzed the expression of FPR2 in PD-MSCs by immunofluorescence. As shown in Figure 1L, FPR2 was expressed in naïve PD-MSCs and was decreased by siRNA-FPR2 transfection, while it was significantly increased by WKYMVm treatment of PD-MSCs (*p* < 0.05, Figure 1K,L). To investigate the effect of WKYMVm on PD-MSCs, we performed ELISA and gelatin zymography with PD-MSC culture supernatant. The expression of VEGF and hepatocyte growth factor (HGF) were significantly upregulated in the WKYMVm-treated PD-MSCs compared to the untreated PD-MSCs and the siRNA-FPR2-transfected PD-MSCs (*p* < 0.05, Figure 1M,N). However, transforming growth factor (TGF)-β was significantly decreased, while the activity of MMP-9 was increased in the WKYMVm-treated PD-MSC group (*p* < 0.05, Figure 1O,P). Therefore, these data demonstrate that WKYMVm enhanced the proangiogenic, regenerative, or antifibrotic effects of PD-MSCs via FPR2.

### 3.3. PD-MSCs Combined with WKYMVm Enhance Hepatic Function in the BDL Rat Model

For analysis of the therapeutic effect of PD-MSCs combined with WKYMVm in a liver-injured rat model, we determined the serum levels of hepatic function markers (e.g., AST, ALT, total bilirubin, and ALB). The BDL rat group (NTx) had significantly increased levels of AST, ALT, and total bilirubin compared to the control. However, PD-MSC transplantation (Tx) slightly reduced the levels of ALT, AST, and total bilirubin. Interestingly, the levels in the PD-MSCs combined with the WKYMVm (Tx+WK) group were evidently lower than those in the NTx and Tx groups (*p* < 0.05, Appendix A). Additionally, the serum level of ALB was significantly upregulated in the Tx+WK group compared with the NTx and Tx groups (*p* < 0.05, Appendix A). Therefore, these results suggest that the combined transplantation of PD-MSCs with WKYMVm improves hepatic function in the BDL rat model.

### 3.4. PD-MSCs Combined with WKYMVm Promote Vascular Regeneration in the BDL Rat Liver

To evaluate the effect of PD-MSCs combined with WKYMVm on hepatic angiogenesis, we analyzed histological changes in portal tracks in liver tissues by H&E staining and the expression levels of angiogenic factors. As shown in Figure 2A, the liver tissues from the NTx group exhibited irregular portal tracts and an increased diameter of the portal vein compared with those of the control group. In contrast, the Tx+WK group showed recovery of the shape of the portal tracts and a decrease in the diameter of the portal vein compared with the NTx and Tx groups (*p* < 0.05, Figure 2A,B). The concentration of VEGF was lower in the NTx group than in the control but significantly increased in the Tx+WK group, which had a much higher level than the Tx group (*p* < 0.05, Figure 2C). The mRNA expression of VEGF in the Tx+WK group was dramatically increased compared to that in the NTx group (*p* < 0.05, Figure 2D). The levels of VEGFR1 and VEGFR2 showed an increasing tendency in the Tx and Tx+WK groups compared to the NTx group (*p* < 0.05, Figure 2E,F).

Additionally, the level of endoglin was significantly upregulated in the Tx+WK group compared to the NTx group (*p* < 0.05, Figure 2G). Furthermore, we examined the expression of the fibrotic marker alpha-smooth muscle actin (α-SMA) and the vascular marker vWF in liver sections using an immunofluorescence assay at 2 weeks after transplantation. Compared with the NTx and Tx groups, the Tx+WK group showed substantially decreased expression of α-SMA and increased vWF expression (*p* < 0.05, Figure 2H–J). These data indicate that PD-MSCs combined with WKYMVm promote vascular regeneration through activation of angiogenic factors in the BDL rat model.

### 3.5. PD-MSCs Combined with WKYMVm Promote Angiogenic Activity in HUVECs

To further determine the angiogenic effect of PD-MSCs combined with WKYMVm in vitro, we used 5-FU as a functional inhibitor of endothelial cells and cocultured the cells with PD-MSCs and WKYMVm, as shown in Figure 3A. The viability of the 5-FU-treated HUVECs was strongly decreased compared with that of the control, whereas the cells cocultured with PD-MSCs alone or PD-MSCs with WKYMVm showed a significant increase (*p* < 0.05, Figure 3B). The concentration of VEGF secreted by HUVECs was dramatically increased in the cells cocultured with PD-MSCs and WKYMVm compared to the 5-FU-treated HUVECs and those cocultured with PD-MSCs. Moreover, the VEGF level showed an increase greater than that of the control (*p* < 0.05, Figure 3C). Previous studies have shown that the canonical Wnt/β-catenin signaling pathway plays a regulatory role in vascular regeneration [32]. Thus, we examined whether PD-MSCs and WKYMVm affect the expression of β-catenin in HUVECs. Immunofluorescence showed that reduced active β-catenin expression in the 5-FU-treated HUVECs was significantly enhanced by PD-MSCs and WKYMVm compared with the control (*p* < 0.05, Figure 3D,E).

Additionally, the protein level of β-catenin in the nucleus was significantly upregulated in the 5-FU-treated HUVECs cocultured with PD-MSCs and WKYMVm versus the HUVECs with only 5-FU treatment (*p* < 0.05, Figure 3F). Tube formation by HUVECs was significantly increased by PD-MSCs and WKYMVm treatment (*p* < 0.05, Figure 3G,H). However, the expression of TGF-β, which is the key factor of endothelial–mesenchymal transition (End-MT), was significantly decreased by PD-MSCs and WKYMVm compared to that of the HUVECs with 5-FU treatment alone (*p* < 0.05, Figure 3I). To examine the effect of PD-MSCs and WKYMVm on endothelial permeability, we performed a dextran permeability assay, as shown in Figure 3J. PD-MSCs alone and PD-MSCs combined with WKYMVm induced a significant decrease in the dextran permeability of the 5-FU-treated HUVECs (*p* < 0.05, Figure 3K). Taken together, these results suggest that coculture with PD-MSCs and WKYMVm promotes angiogenic activities of ECs by upregulating the expression of VEGF and β-catenin.

### 3.6. PD-MSCs Combined with WKYMVm Attenuate Hepatic Fibrosis in the BDL Rat Liver

The effect of PD-MSCs combined with WKYMVm against BDL-induced hepatic fibrosis was evaluated through Sirius Red. As shown in Figure 4A, extensive accumulation of collagen was observed in the liver tissues of the NTx group. In contrast, accumulated collagen was shown to decrease in the Tx group. Additionally, collagen deposition was evidently reduced in the Tx+WK group compared to the NTx group (*p* < 0.05, Figure 4A,B). To verify the antifibrotic effect of the Tx and Tx+WK groups, we examined the expression levels of epithelial markers (e.g., E-cadherin) and mesenchymal and fibrogenic markers (e.g., vimentin, Col I, and α-SMA) at the protein level. As shown by E-cadherin expression, the control maintained basal levels. However, in the Tx and Tx+WK groups, the level of E-cadherin was dramatically increased compared with that in the NTx group (*p* < 0.05, Figure 4C).

In contrast to the expression of E-cadherin, the expression of vimentin and α-SMA showed a decreasing tendency in the Tx group and a significant decrease in the Tx+WK group (*p* < 0.05, Figure 4D,F). In particular, Col I was evidently downregulated in the Tx+WK group compared to the NTx and Tx groups (*p* < 0.05, Figure 4E). These results indicate that administration of PD-MSCs combined with WKYMVm alleviates hepatic fibrosis by upregulating the expression of epithelial markers and inhibiting the expression of mesenchymal markers in a BDL rat model.

### 3.7. PD-MSCs Combined with WKYMVm Inhibit HSC Activation In Vitro

To further investigate the antifibrotic effect of PD-MSCs and WKYMVm on hepatic stellate cells (HSCs), we cocultured HSCs activated by TGF-β treatment with PD-MSCs and WKYMVm, as shown in a Figure 5A. The mRNA and protein levels of Col I and α-SMA in the TGF-β-treated HSCs were substantially upregulated compared to those of the control, whereas these levels were significantly downregulated by cotreatment with PD-MSCs alone or PD-MSCs and WKYMVm, and PD-MSCs and WKYMVm had greater inhibitory effects (*p* < 0.05, Figure 5B–E). In addition, immunofluorescence data showed that cotreatment with PD-MSCs and WKYMVm significantly reduced α-SMA expression compared to the other treatments (*p* < 0.05, Figure 5F,G). Matrix metalloproteinases (MMPs) play crucial roles in the regulation of hepatic fibrosis and regeneration. MMP2 and MMP9 specialize in degrading extracellular matrices secreted from activated HSCs [33]. Through gelatin zymography, the activities of MMP2 and MMP9 were shown to be reduced in the TGF-β-treated HSCs compared to the control, but the activities were elevated by cotreatment of PD-MSCs and WKYMVm (*p* < 0.05, Figure 5H,I). These results suggest that treatment with PD-MSCs combined with WKYMVm inhibits fibrogenesis of activated HSCs.

### 3.8. PD-MSCs Combined with WKYMVm Improve Hepatic Regeneration in the BDL Rat Model

Hepatocyte nuclear factor 1 alpha (HNF1α) is a transcription factor that regulates the expression of several liver-specific genes [34]. The mRNA level of HNF1α was slightly increased in the Tx and Tx+WK groups compared with the NTx group (Figure 6A). The mRNA and protein levels of ALB were significantly increased in the Tx+WK group compared with the NTx and Tx groups (*p* < 0.05, Figure 6B,E). Additionally, the receptors of the WKYMVm peptide are FPR2 and HGF receptors (HGFR; tyrosine-protein kinase Met, c-Met) [35]. The mRNA levels of FPR2 and HGFR in rat liver were maintained at basal levels. In the NTx group, their expression levels showed a decreasing tendency compared with those of the control. However, PD-MSCs with WKYMVm resulted in significantly increased expression compared to that in the NTx group (*p* < 0.05, Figure 6C,D).

Hepatic regeneration is initiated by several growth factors and the IL-6/gp130/STAT3 signaling pathway. To determine whether the combination therapy of PD-MSCs with WKYMVm alleviates hepatic damage in the BDL model, we analyzed the expression of factors related to liver regeneration using Western blotting. We examined the protein levels of IL-6, gp130 and the phosphorylated form of STAT3 in the BDL rat liver. Their expression levels were decreased in the NTx group but gradually increased in the Tx and Tx+WK groups (Figure 6E). The serum level of HGF was also significantly increased in the Tx+WK group compared to the NTx group (*p* < 0.05, Figure 6H). To further examine the effect of the combination therapy of PD-MSCs with WKYMVm on hepatocyte proliferation, we assessed PCNA expression in liver tissues by immunohistochemistry and cyclin D1 expression by Western blot. Few PCNA-positive hepatocytes were observed in the NTx group, while a significant increase in PCNA-positive cells was found in the Tx+WK group compared to the NTx and Tx groups (*p* < 0.05, Figure 6F,G). Additionally, the level of cyclin D1 was upregulated in the Tx+WK group (*p* < 0.05, Figure 6E). These findings suggest that combination therapy with PD-MSCs and WKYMVm in the rats with BDL promotes hepatic regeneration through induction of hepatocyte-specific proteins and the IL-6/gp130/STAT3 signaling pathway.

### 3.9. PD-MSCs Combined with WKYMVm Can Regenerate Damaged Hepatocytes In Vitro

To further analyze their mode of action in vitro, we performed a coculture experiment of LCA-treated WB-F344 rat liver epithelial cells with PD-MSCs and WKYMVm, as shown in Figure 7A. Cell proliferation was measured using a CCK-8 assay. The CCK-8 assay demonstrated that treatment of WB-F344 cells with PD-MSCs and WKYMVm significantly increased proliferation (*p* < 0.05, Figure 7B). Treatment of the LCA-treated WB-F344 cells with PD-MSCs and WKYMVm strongly enhanced the protein expression of ALB and HNF1α, as shown by Western blot, compared to that of the LCA-treated group (*p* < 0.05, Figure 7C,D). Moreover, immunofluorescence data showed downregulated HNF1α expression in the LCA-treated WB-F344. However, HNF1α expression was significantly upregulated by treatment with PD-MSCs and WKYMVm, similar to the control (*p* < 0.05, Figure 7E,F). These data indicate that cotreatment with PD-MSCs and WKYMVm may promote the repair and regeneration of injured hepatocytes.

## 4. Discussion

A disruption in angiogenesis contributes to numerous ischemic, inflammatory, immune, and malignant disorders [36]. Upon injury to cells or tissues, inflammation or hypoxia results in the generation of angiogenic mediators that regulate the migration of vascular precursor cells from their niche to the site of injury [37]. In our study, we first demonstrated that the combination of PD-MSCs and WKYMVm has proangiogenic and antifibrotic effects and therapeutic effectiveness in a BDL-injured rat model. PD-MSCs and WKYMVm decreased the levels of AST, ALT, and total bilirubin while increasing ALB and hepatokines levels, which are important indices of hepatic function (*p* < 0.05, Appendix A). Additionally, PD-MSCs with WKYMVm improved hepatic architectural distortion and collagen deposition in fibrotic liver tissues (*p* < 0.05, Figure 2A and Figure 4A).

During hepatic injury, HSCs activated by TGF-β upregulate the expression of Col I and α-SMA. The NTx group showed dramatically activated HSCs, as demonstrated by the increased expression of vimentin, Col I, and α-SMA in Western blots, while the Tx and Tx+WK groups displayed significantly downregulated expression levels of these molecules (*p* < 0.05, Figure 4D–F). In vitro, PD-MSCs and WKYMVm repressed the expression of α-SMA and Col I in the rat HSCs, which underwent trans-differentiation by TGF-β (*p* < 0.05, Figure 5B–G). Our in vivo and in vitro studies showed that the combination therapy of PD-MSCs and WKYMVm has an antifibrotic effect in the liver.

In hepatic regeneration, vascular remodeling plays an important role in many physiological and pathological events [38,39]. HSCs and ECs can promote physiological angiogenesis by expressing a variety of angiogenic factors [40]. VEGF is a key regulator of angiogenesis and vascular regeneration and is also known as an essential element for EC viability. In partial hepatectomy or acute hepatic failure, VEGF is evidently increased, and sinusoidal reconstruction is improved during liver regeneration [5,41]. Yang and his colleagues found that VEGF has a dual and opposing role in fibrogenesis and resolution of fibrosis through critical effects of VEGF on vascular permeability [8]. In ischemic heart disease, WKYMVm had proangiogenic effects via VEGF signaling [26]. Additionally, in our study, PD-MSCs with WKYMVm significantly induced vascular regeneration in vivo and in vitro, as demonstrated by the increased expression of angiogenic factors and tube formation and decreased dextran permeability (*p* < 0.05, Figure 2 and Figure 3).

Recently, ECs were shown to differentiate into myofibroblasts through EndMT in the lung, kidney, and heart [42,43,44]. Ribera and his colleagues showed the mesenchymal phenotype of ECs in CCl_4_-induced liver cirrhosis via bone morphogenic protein-7 (BMP-7)/TGF-β signaling [45]. Dufton et al. showed that the ETS-related gene (ERG), a transcription factor in the endothelial lineage, repressed EndMT in a CCl4-injured cirrhotic mouse model. Additionally, ERG attenuated endothelial-dependent hepatic fibrosis by inhibiting SMAD2/3 signaling while maintaining hepatic homeostasis by inducing the SMAD1 pathway [46]. We also demonstrated that the levels of mesenchymal markers, such as vimentin, Col I, and α-SMA, were dramatically decreased in the PD-MSC-transplanted group and in the group treated with PD-MSCs and WKYMVm in cirrhotic rats with BDL (*p* < 0.05, Figure 4).

Park et al. reported that WKYMVm decreases dermal thickness and inhibits scleroderma fibrosis in bleomycin-induced mice by repressing vimentin and phosphorylated SMAD3 expression in myofibroblasts [47]. Additionally, the peptide was demonstrated to inhibit osteoclastogenesis by decreasing the levels of inflammatory cytokines such as IL-1β and tumor necrosis factor-alpha (TNF-α) via the CD9/gp130/STAT3 pathway [48]. However, WKYMVm has a short half-life. Park et al. reported that the anti-inflammatory effect of peptides is constrained since high levels are required for therapeutic efficacy [49]. Therefore, other studies sought to address the limitation of WKYMVm. Additionally, PLGA microspheres were used to overcome the short half-life of WKYMVm [29]. To extend the short elimination half-life of WKYMVm, we developed a combination therapy of PD-MSCs with WKYMVm. For therapy using MSCs, other medications or functional genes have been used to enhance the function of MSCs.

## 5. Conclusions

The administration of PD-MSCs with WKYMVm peptide improves vascular regeneration by activated FPR2 signaling while attenuating hepatic fibrosis in a rat model of liver cirrhosis. Furthermore, liver regeneration is promoted by PD-MSCs with WKYMVm by increasing the expression of hepatic function markers. Therefore, this study suggests the strong therapeutic efficacy of PD-MSCs combined with WKYMVm as a regulator of hepatic function as well as vascular regeneration in hepatic failure.

## Figures and Tables

**Figure 1 cells-11-00232-f001:**
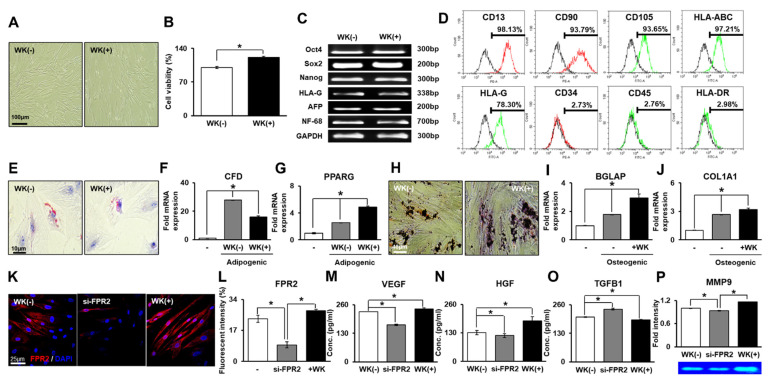
WKYMVm treatment maintains and enhances PD-MSC characteristics and effects. (**A**) Morphology of the PD-MSCs and WKYMVm-treated PD-MSCs. Scale bar = 100 μm. (**B**) Viability of the PD-MSCs and WKYMVm-treated PD-MSCs shown by CCK-8 assays. (**C**) Expression of stemness-related markers in the WKYMVm-treated PD-MSCs shown by RT-PCR. (**D**) MSC surface markers (e.g., hematopoietic, nonhematopoietic, and HLA family) in the WKYMVm-treated PD-MSCs using FACS analysis. (**E**) Lipid droplets in PD-MSCs differentiated into adipocytes shown by Oil Red O staining. Scale bar = 10 μm. Expression of adipogenic-specific markers such as CFD (**F**) and PPARG (**G**) shown by qRT-PCR. (**H**) Calcium deposition of PD-MSCs differentiated into osteocytes shown by von Kossa staining. Scale bar = 40 μm. Expression of osteogenic-specific markers, including BGLAP (**I**) and COL1A1 (**J**), shown through qRT-PCR. (**K**) Expression of FPR2 in PD-MSCs using immunofluorescence. Scale bar = 25 μm. (**L**) Quantification of the fluorescence intensity of FPR2 through immunofluorescence. Expression of VEGF (**M**), HGF (**N**), and TGFB1 (**O**) in the WKYMVm-treated PD-MSC culture supernatants shown by ELISAs. (**P**) The activity of MMP9 in the WKYMVm-treated PD-MSC culture supernatant shown using gelatin zymography. Mean ± SD, * *p* < *0*.05 by *t* tests. si-FPR2, siRNA-FPR2-transfected PD-MSCs; +WK, WKYMVm-treated PD-MSCs.

**Figure 2 cells-11-00232-f002:**
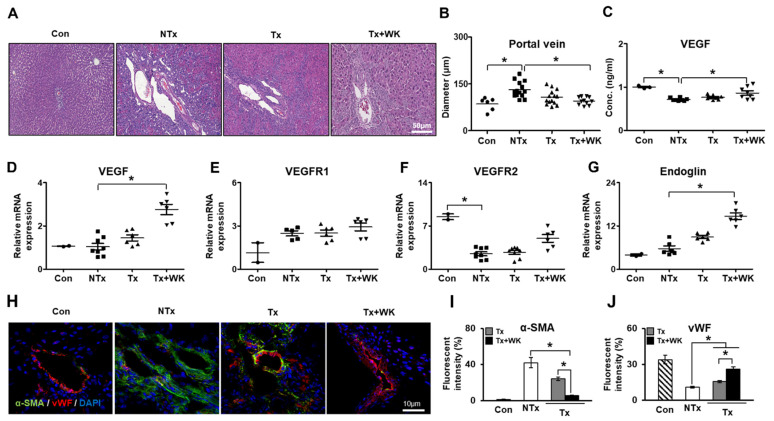
PD-MSCs combined with WKYMVm promoted vascular regeneration in the BDL rat liver. (**A**) Histological phenotype of BDL rat liver by H&E staining. Scale bar = 50 μm. (**B**) Quantification of portal vein diameter in BDL rat liver through H&E staining. (**C**) VEGF level in BDL rat serum shown by ELISAs. mRNA expression of VEGF (**D**), VEGFR1 (**E**), VEGFR2 (**F**), and CD105 (**G**) shown by qRT-PCR. (**H**) Expression of α-SMA and vWF in BDL rat livers by immunofluorescence. Scale bar = 10 μm. Quantified fluorescence intensity of α-SMA (**I**) and vWF (**J**) through immunofluorescence. *n* = 3 rats per group, mean ± SD, * *p* < 0.05 by one-way ANOVA. Con, control; NTx, nontransplantation group; Tx, PD-MSC transplantation group; Tx+WK, PD-MSCs with WKYMVm combined transplantation group; wk, week.

**Figure 3 cells-11-00232-f003:**
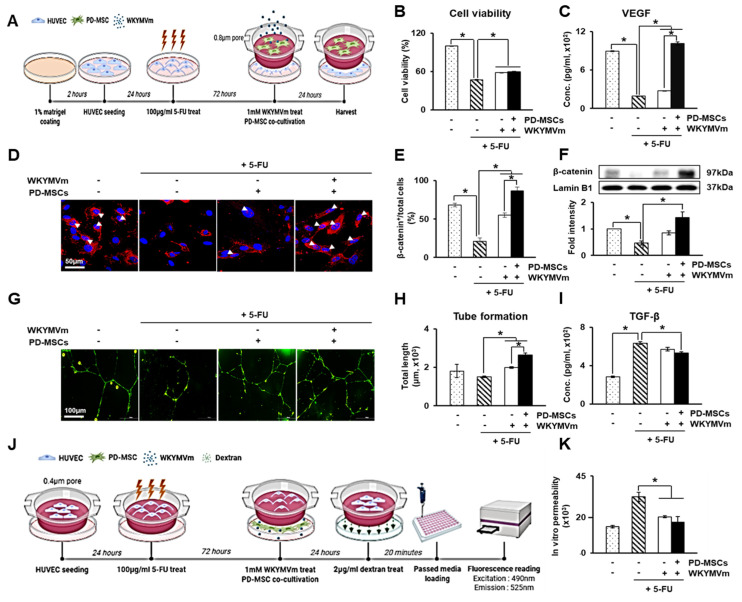
PD-MSCs combined with WKYMVm promote angiogenic activity in HUVECs. (**A**) In vitro scheme in HUVECs. (**B**) Cell viability in the 5-FU-treated HUVECs determined by CCK-8 assays. (**C**) VEGF level in the culture supernatant of HUVECs treated with 5-FU shown by ELISAs. (**D**) Localization and expression of active β-catenin through immunofluorescence. Scale bar = 50 μm. (**E**) Quantification of active β-catenin-positive HUVECs versus total HUVECs using immunofluorescence. (**F**) The protein expression of active β-catenin in the nucleus of HUVECs. (**G**) Tube formation of HUVECs injured by 5-FU. Scale bar = 100 μm. (**H**) Quantification of total tube length through tube formation assays. (**I**) TGF-β levels in the culture supernatant of HUVECs treated with 5-FU shown by ELISAs. (**J**) Schematic figure of the dextran permeability assay in the HUVECs treated with 5-FU. (**K**) Dextran permeability of the HUVECs injured by 5-FU. *n* = 3 per group, mean ± SD, * *p* < 0.05 by *t* tests.

**Figure 4 cells-11-00232-f004:**
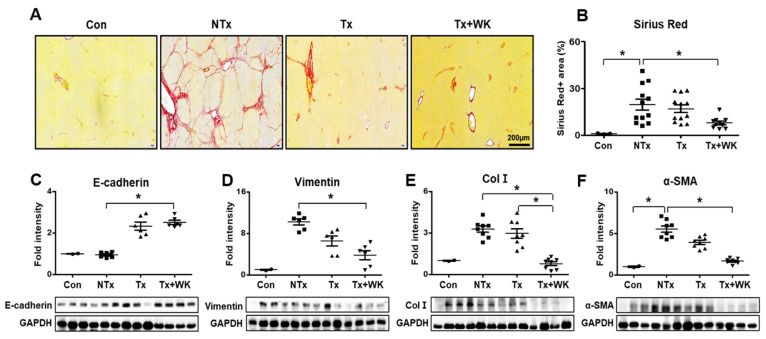
PD-MSCs combined with WKYMVm attenuated hepatic fibrosis in the BDL rat liver. (**A**) Collagen deposition in BDL rat liver shown by Sirius Red staining. Scale bar = 200 μm. (**B**) Quantification of the Sirius Red-positive area in BDL rat livers through Sirius red staining. Protein expression of E-cadherin (**C**), vimentin (**D**), Col I (**E**), and α-SMA (**F**) in BDL rat livers shown by Western blots. Protein expression was normalized to GAPDH expression through Western blot bands. *n* = 3 rats per group, mean ± SD, * *p* < 0.05 by one-way ANOVA. Con, control; NTx, nontransplantation group; Tx, PD-MSC transplantation group; Tx+WK, PD-MSCs with WKYMVm combined transplantation group; wk, week.

**Figure 5 cells-11-00232-f005:**
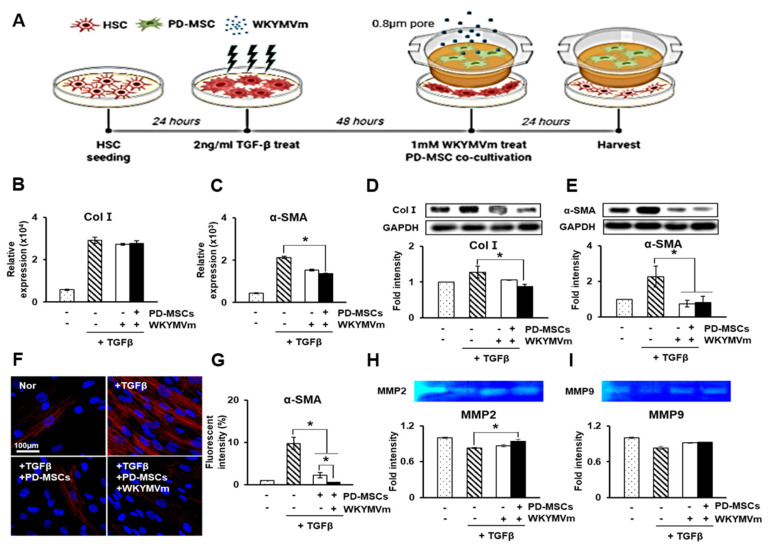
PD-MSCs combined with WKYMVm inhibit HSC activation in vitro. (**A**) Schematic figure of HSC in vitro modeling. mRNA expression of Col I (**B**) and α-SMA (**C**) in HSCs treated with TGF-β shown by qRT-PCR. Protein levels of Col I (**D**) and α-SMA (**E**) shown by Western blots. (**F**) α-SMA expression in HSCs treated with TGF-β, as determined by immunofluorescence. Scale bar = 100 μm. (**G**) Fluorescence intensity of α-SMA, as determined by immunofluorescence. Enzyme activity of MMP-2 (**H**) and MMP-9 (**I**) in the culture supernatant of HSCs treated with TGF-β shown by gelatin zymography. *n* = 3 per group, mean ± SD, * *p* < 0.05 by *t* tests.

**Figure 6 cells-11-00232-f006:**
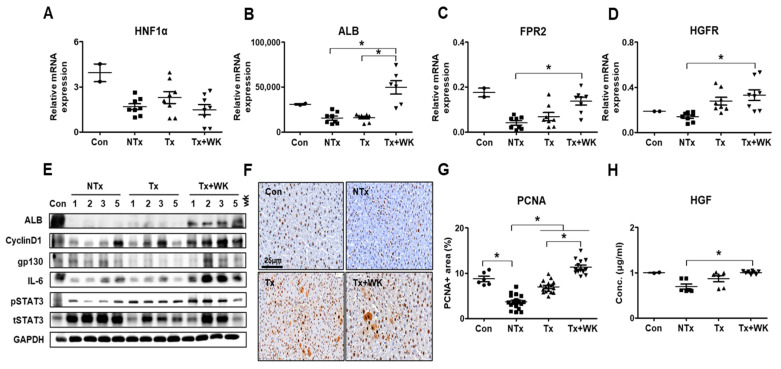
PD-MSCs combined with WKYMVm enhance hepatic regeneration in the BDL rat model. mRNA expression of HNF1α (**A**), ALB (**B**), FPR2 (**C**), and HGFR (**D**) in BDL rat liver shown by qRT-PCR. (**E**) Protein levels of ALB, cyclin D1, gp130, IL-6, p/tSTAT3, and GAPDH in BDL rat livers shown by Western blots. (**F**) Expression of PCNA in BDL rat liver shown by immunohistochemistry. Scale bar = 25 μm. (**G**) Quantification of the PCNA-positive area in BDL rat livers through immunohistochemistry. (**H**) Secreted HGF level in BDL rat serum shown by ELISAs. *n* = 3 rats per group, mean ± SD, * *p* < 0.05 by one-way ANOVA. Con, control; NTx, nontransplantation group; Tx, PD-MSC transplantation group; Tx+WK, PD-MSCs with WKYMVm combined transplantation group; wk, week.

**Figure 7 cells-11-00232-f007:**
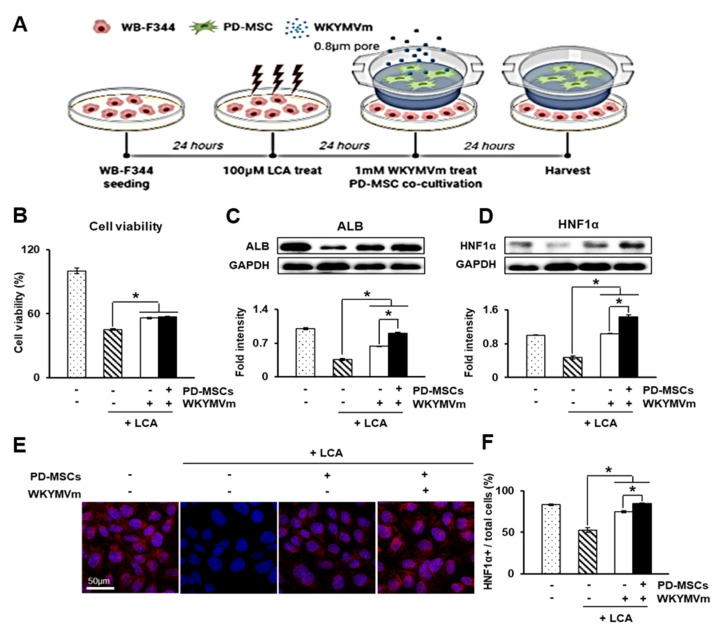
PD-MSCs combined with WKYMVm can regenerate damaged hepatocytes in vitro. (**A**) Schematic figure of WB-F344s cell in vitro modeling. (**B**) Cell viability of the LCA-treated WB-F344s cells shown by CCK-8 assays. Protein expression of ALB (**C**) and HNF1α (**D**) shown using Western blots. Protein expression was normalized to GAPDH expression through Western blot bands. (**E**) Localization and expression of translocated HNF1α in the nucleus shown through immunofluorescence. (**F**) Quantification of translocated HNF1α-positive WB-F344s versus total WB-F344s using immunofluorescence. *n* = 3 per group, mean ± SD, * *p* < 0.05 by *t* tests.

## Data Availability

Not applicable.

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
