# Peer review of "Combination Therapy of Placenta-Derived Mesenchymal Stem Cells with WKYMVm Promotes Hepatic Function in a Rat Model with Hepatic Disease via Vascular Remodeling"

_cells, 2022, doi:10.3390/cells11020232_

Round 1

Reviewer 1 Report

This investigation was addressed to determine the combined effect of placenta-derived mesenchymal stem cells with WKYMVm peptide on vascular remodeling during liver alteration induced by bile duct ligation, such as fibrogenesis, angiogenesis, and regeneration. Authors generated interesting in vitro and in vivo evidence showing that the above combination decreases collagen accumulation, and improve liver function by increasing hepatocyte proliferation and albumin, as well as, increases angiogenesis by activated FPR2 signaling. Also, they propose that combined effect of PD-MSCs with WKYMVm peptide might be an attractive therapeutic strategy for liver diseases. Nonetheless, this version of the manuscript still needs important improvements before proceeding further.

The title of the investigation is confusing since it states that combination of PD-MSCs with WKYMVm peptide “accelerates” hepatic disease via vascular remodeling; however, both in vitro and in vivo evidence shows that those combination regresses the liver alteration induced by liver injury models. So, title must be carefully rewritten.

The manuscript contains a number of syntax issues, so, it needs a thorough and meticulous revision by a native English speaker.

Based on HUGO Gene Nomenclature Committee, gene and protein symbols must be indicated according to the accepted nomenclatures. So, throughout the manuscript including abstract, main text, figures, figures legends, and supplementary material, gene and protein symbols must be properly indicated. Authors should know that both gene and protein symbols are different among species. As reference, authors should review HUGO website as well as the following article: PMID: 22836666. For example, gene expression plots in figure 1 and supplementary Table 1 do not properly cite gene symbols. Authors should be aware that both gene and protein short names are symbols, not abbreviations. Please, carefully review the whole manuscript.

The authors included some experiments associated with cellular senescence; however, those experiments seem like a patch in the research. It seems that authors forced the inclusion of that experiments since no associated antecedents were included in introduction section, but more importantly, they did not discuss anything about the role of senescence in vascular remodeling, as well as, the effect of PD-MSCs with Peptide WKYMVm on senescence.

Authors included some cell senescence-associated experiments; however, it seems that those experiments were forced to be included since no associated background was included in introduction section but more important, authors did not discuss anything about the role of senescence in vascular remodeling, as well as, the effect of PD-MSCs with WKYMVm peptide on senescence.

In result section, authors must include real p values instead of the accepted lower limit (p <0.05) only. We all are aware that p value must be under 0.05 for considering a statistical difference as significative; so, to trust in shown results, readers need to know the real calculated p value.

For an easier reproducibility of the experiment by other researcher, catalog number of antibodies and that of all others reagents should be included.

What does mean 23592412 and 12454508, on Pages 2 and 14, lines 56 and 488, respectively? Moreover, what does mean 30 on Page 3, line 101? It seems that authors were not careful in the review of the final version of the manuscript.

Author Response

January 5, 2022

Cover letter

Article Submission in Cells    

Dear, Dr. Editor,

Editor-in-Chief

We greatly appreciate your careful evaluation of our manuscript (Cells-1530895) entitled “Combination therapy of placenta-derived mesenchymal stem cells with WKYMVm accelerates hepatic function in hepatic disease via vascular remodeling.” We were really encouraged by the reviewers’ positive comments and constructive suggestions.

In revision, we are more carefully revised all issues and concerns through additional data and subsequent revision of our manuscript, as detailed in the following response page. As Reviewer’s commented, we corrected it clearly stating with each comment and changes are highlighted in red in the revised manuscript. In summary, based on the insightful and constructive criticisms provided by referees, we believe that our manuscript is significantly improved and we hope that you will consider it suitable for publication in Cells.

With warm personal regards,

Gi Jin Kim, Ph.D.

Associate Professor

Dept. of Biomedical Science, CHA University;

689, Sampyeong-dong, Bundang-gu, Seongnam-si, Gyeonggi-do, Republic of Korea.

Tel: 82-31-881-3687, Fax: 82-31-881-4102, e-mail: [email protected]

Reviewer 2 Report

This study is very important because the authors first to show that the combination of PD-MSCs and WKYMVm has proangiogenic and antifibrotic effects and therapeutic effectiveness in a surgical rat model of liver fibrosis (BDL). 

The research in this study is very complex and has been performed in both in vivo (BDL-rat model of liver fibrosis) and in vitro studies (cell cultures).

The experimental protocols were approved by the Institutional Animal Care and Use Committee of CHA University,  Seongnam, Korea (IACUC-180023).

The results are clear and in detail shown in the figures.

Comments:

In paragraph 2. Materials and Methods (2.4) instead Serological analysis put Biochemical analysis

Is there any data on the effect of PD-MSCs and WKYMVm on mechanisms of autophagy and apoptosis in liver fibrosis?

Author Response

Dear, Dr. Editor,

Editor-in-Chief

We greatly appreciate your careful evaluation of our manuscript (Cells-1530895) entitled “Combination therapy of placenta-derived mesenchymal stem cells with WKYMVm accelerates hepatic function in hepatic disease via vascular remodeling.” We were really encouraged by the reviewers’ positive comments and constructive suggestions.

In revision, we are more carefully revised all issues and concerns through additional data and subsequent revision of our manuscript, as detailed in the following response page. As Reviewer’s commented, we corrected it clearly stating with each comment and changes are highlighted in red in the revised manuscript. In summary, based on the insightful and constructive criticisms provided by referees, we believe that our manuscript is significantly improved and we hope that you will consider it suitable for publication in Cells.

With warm personal regards,

Gi Jin Kim, Ph.D.

Associate Professor

Dept. of Biomedical Science, CHA University;

689, Sampyeong-dong, Bundang-gu, Seongnam-si, Gyeonggi-do, Republic of Korea.

Tel: 82-31-881-3687, Fax: 82-31-881-4102, e-mail: [email protected]

Reviewer 3 Report

Point 1: In figure 1M, did the authors check the knockdown efficiency of FPR2 siRNA by Western blotting in addition to immunofulorescence?

Point 2: In figures 2A and 2B, there was a little bit of different trend between the representative image (2A) and quantitative results shown in 2B. 

Point 3: Is it significant for the changes of VEGFR1 in figure 2E?

Point 4: In figure 4C to 4F, there was no comparable trend in the representative Western blotting and the quantitative results. 

Point 5: For the quantitative RT-PCR, it seems no standard control sample used and each fold change used in each data in figure 6 was different. 

Author Response

(The authors gave the same response as above.)
